# Effects of Vojta Therapy on the Motor Function of Children with Neuromotor Disorders: Study Protocol for a Randomized Controlled Trial

**DOI:** 10.3390/jcm12237373

**Published:** 2023-11-28

**Authors:** Mónica Menéndez-Pardiñas, Miguel Alonso-Bidegaín, Fernando Santonja-Medina, Juan Luis Sánchez-González, Jose Manuel Sanz-Mengibar

**Affiliations:** 1Unidad de Atención Temprana y Rehabilitación Infantil, del Servicio de Medicina Física y Rehabilitación, del Hospital Materno Infantil Teresa Herrera, del Área Sanitaria de A Coruña y Cee, 15006 A Coruña, Spain; monicamenendez31@gmail.com (M.M.-P.); miguel.alonso.bidegain@sergas.es (M.A.-B.); 2Facultad de Fisioterapia, Universidade da Coruña, 15471 Ferrol, Spain; 3Faculty of Medicine, University of Murcia, 30100 Murcia, Spain; fernando@santonjatrauma.es; 4Department of Traumatology, V. de la Arrixaca University Hospital, 30120 Murcia, Spain; 5Department of Nursing and Physiotherapy, Faculty of Nursing and Physiotherapy, Instituto de Investigación Biomédica de Salamanca (IBSAL), Campus Miguel de Unamuno s/n, 37007 Salamanca, Spain; 6Centre for Neuromuscular Diseases, National Hospital for Neurology and Neurosurgery, University College London Queen Square, London WC1N 3BG, UK; j.sanzmengibar@ucl.ac.uk

**Keywords:** cerebral palsy, Vojta therapy, neuromotor disorder, neurorehabilitation, brain injury

## Abstract

Background: Infantile cerebral palsy is a neurological pathology that causes great morbidity, mortality, and disability in people who suffer from it, mainly affecting motor development. There are a multitude of non-pharmacological methods or therapies for its treatment. One of the main methods is Vojta therapy. This methodology acts on ontogenetic postural function and automatic postural control. Objective: This study aims to demonstrate that there are changes in the motor development of children with cerebral palsy with the application of Vojta therapy. Methods and analysis: This is a randomized controlled trial on the effectiveness of two neurorehabilitation techniques in patients with cerebral palsy conducted at the Physical Medicine and Rehabilitation Service of the Teresa Herrera Maternal and Child Hospital of the A Coruña and Cee Health Area. The study will be conducted from January 2023 to December 2024. There will be two groups: the Vojta therapy group (*n* = 30) and the conventional physiotherapy group (*n* = 30). The measurement variables will be gross motor function as measured by the Gross Motor Function Measure (GMFM) and Infant Motor Profile (IMP) scales. Ethics and dissemination: The study was approved by the Research Ethics Committee of the University of Murcia (1823/2018) and Comité de Ética de la Investigación de A Coruña-Ferrol (2022/099). Trial registration number: ClinicalTrials.gov; identifier: NCT06092619.

## 1. Introduction

Considering the time and effort that families of children with cerebral palsy invest in physical therapy, it is important to know whether more frequent interventions really improve gross motor performance. Some recent studies suggested that high-frequency physical therapy should be considered when gross motor improvement is prioritized [1,2,3]. Motor learning theories highlight the need for high-frequency repetitions for motor learning to occur. Due to these recommendations, most children with cerebral palsy are immersed in goal-directed, task-specific practice with frequent repetitions. Nevertheless, multiple environmental factors [4] may affect gross motor progress, including how everyday skills may be integrated into daily routines, thereby providing additional therapy and making it difficult to understand how much progress come directly for physical therapy intervention [5]. On the other hand, Vojta therapy (VT) was developed by Professor Vojta and uses locomotor patterns activated “reflexogenically”. This methodology acts on the ontogenetic postural function and automatic postural control [6], on which different environmental aspects will later act. Also known as “Reflex Locomotion”, it does not refer to neuronal regulation but is rather related to therapeutically applied external stimuli and their automatic movement responses. VT was developed by the Czech professor and pediatric and adult neurologist Vaclav Vojta in the 1960s. Most of his rehabilitation work and the development of the Vojta principle was developed in Germany, where he emigrated due to the political situation in Europe after the Second World War. The physiotherapy developed by Professor Vojta using “reflex locomotion” is based on his findings on the development of posture and movement in children. He discovered the basis of this therapeutic intervention while searching for treatment possibilities for young people with CP and spastic paralysis [7]. His approach was founded in years of systematic neurokinesiological observation and examination of children with typical and atypical development. He observed that sustained stimulation of peripheral pressure elicited a stereotyped full-body motor response, manifested as a pattern of tonic and phasic muscle contractions on both sides of the neck, trunk, and limbs, resulting from spatial and temporal summation that leads to improved postural control [6]. Therapists administer pressure to defined zones on the body in a patient who is in a prone, supine, or side-lying position. Two main movement complexes are used therapeutically: “reflex creeping” in a prone lying position and “reflex rolling” from a supine and side-lying position [7,8], The afferent input of VT therapy generates important cortical and subcortical changes, especially in the ipsilateral putamen [9]. The resulting outputs are differential responses that seem to be located in the brainstem and cerebellum, pontomedullary reticular formation, and posterior cerebellar hemisphere and vermis [10], causing muscle contractions in various muscle groups [11]. Studies using EEGs show changes in motor areas, indicating the activation of innate locomotor circuits [12]. These automatic motor patterns aim to change task-related motor activation and its associated postural control, as well as to release and facilitate the development of the ontogenetic postural function [13,14]. Therapeutic components, such as inhibition of abnormal movement patterns, and facilitation of normal movement patterns and physical guidance, have been attributed to VT [15], and are included within the body structure, function, and activity components of the International Classification of Functioning, Disability, and Health (ICF) [16]. Parental education is also a baseline contextual factor within this framework in VT [15].

Children with CP have specific deficits of postural control in standing or walking, including reduced stability of the axial segments. This results in a set of persistent atypical movements and posture, which prevent balance, as well as fluid and selective movements. Compensatory postural strategies described as “on bloc movements” are often observed, such as non-selective swinging between the head and trunk as a single segment to achieve forward propulsion without losing balance [17]. These compensatory strategies must be identified and considered in rehabilitation programs for CP children. These aspects may be improved with rehabilitative interventions focusing on postural control and trunk activity [18,19]. 

For this reason, studies involving intervention with Vojta therapy, which follows these same principles, also seem to demonstrate its effectiveness in these aspects. VT is a commonly extended tool in the field of pediatric rehabilitation [6,20]. Numerous studies have reported improvements in the acquisition of sitting in infantile cerebral palsy [21], gait acquisition [20,22], improving gait kinetics and kinematics [23,24,25], as well as stability, trunk balance, and spinal symmetry [26,27]. Furthermore, Vojta therapy is considered to be one of the most recommended techniques for the treatment of respiratory disorders in infantile cerebral palsy [28]; it has proven to be the technique of choice in respiratory physiotherapy as it has been shown to improve compliance, SO2, and SpO2 by reducing the respiratory frequency without increasing PCO2 [29,30,31]. Furthermore, previous research showed how VT accelerates the acquisition of GMFM-88 items and locomotor stages in children with cerebral palsy younger than 18 months by activating the postural control required to achieve the uncompleted items. This work pointed to the need to develop future studies analyzing the effects of VT at different treatment dosing, with blind assessments and larger samples [6]. Previous studies described significant differences in GMFM sitting after VT, even if the total score of the GMFM-88 did not [21]. VT has been reported to be clinically beneficial for strength, movement, and motor activities in individual cases [32] and is being considered within the second of three levels of evidence in interventions for cerebral palsy [33]. Poor study design has cast a shadow over the positive results in previous studies about VT, including a lack of random sequence generation, concealed allocation, study blinding, incomplete outcome data collection, and selective reporting [33]. On the other hand, a randomized controlled trial showed improvements in postural control and motor function early rehabilitation of adults with acute stroke [34,35]. The ethical limitations of randomized untreated control groups or placebo interventions limit the quality of the intervention in this field. Baseless attributed child distress [33] has been found in the literature [36,37,38], and bonding facilitation due to an integrative approach of the family in the rehabilitation process has also been described [37]. Family participation in the decision-making process is highly emphasized [39], and its integration on the therapeutic handling reduces economic cost and commuting times and increases the dosing possibilities. 

Because functional training of compensatory strategies is not emphasized, gross motor improvements after VT intervention seems to depend on the activation of the postural frame required for their acquisition, reducing compensatory strategies during the subsequent environment exposure. Our hypothesis is that there is an improvement in the postural function and, therefore, on the rate of acquisition of gross motor function from the first weeks of application of reflex locomotion therapy in children with neuromotor disorders. The purposes of this trial were to understand the effect of reflex locomotion therapy on the GMFM of children with cerebral palsy without specific training of its items, as well as to understand what role is played by its dosage. 

### 1.1. Hypothesis

“Vojta or Reflex Locomotion therapy improves the rate of development of gross motor function in babies and children with neuromotor disorders”.

### 1.2. Main Goal

This study aims to demonstrate that there are changes in the motor development of children with cerebral palsy with the application of Vojta therapy by quantifying the impact on the rate of development of gross motor function in children younger than 36 months of age with neuromotor disorders.

### 1.3. Secondary Goals

To understand the effect of both the dosage of hospital treatment and home treatment by parents (economic analysis/resource efficiency).To study the natural history of the rate of development of gross motor function in children under one and a half years of age with neuromotor alterations (pre-rehabilitation intervention).To assess the long-term impact on the gross motor function of children with neuromotor disorders undergoing Vojta therapy compared to children treated without this methodology.

## 2. Materials and Methods

### 2.1. Study Design

A pseudo-randomized clinical trial has been designed that will assess gross motor function before the physiotherapy intervention (with Vojta therapy or conventional physiotherapy intervention, such as environmental enrichment and task-specific training) and two months after its application. The clinical trial has been conducted according to the consolidated standards of reporting trials (CONSORT) statement [40]. The current treatment protocol is described according to the recommendations of SPIRIT [41].

Patients are assigned to one treatment or another following criteria of immediacy as part of the regular routine of the rehabilitation service. Therefore, the randomization of the sample is possible following the natural ordering of the rehabilitation service where the study is carried out. The participants in this research, within their care routine and according to the health protocols established in the service, will be assigned to one of the two study groups:Intervention group: patients under Vojta Therapy intervention [17,18,19,20,21,22,23,24,25,26,27,28]Control group: patients under regular physiotherapy intervention, such as environmental enrichment and task-specific training [25]

This methodology will be useful to provide a service evaluation of the operation of the rehabilitation service itself.

Neither the therapists nor the evaluators of the GMFM may be blinded to the type of treatment administered due to the organization of the rehabilitation service where the study is carried out, as well as the “face-to-face” requirements for carrying out the Vojta therapy and the evaluation of that scale. However, the assessment of the Infant Motor Profile (IMP) is performed by observing a protocolized video recording. This allows the quantification of this factor by an evaluator external to the service (in this case the external principal investigator). This will not only allow us to blind the evaluator regarding the type of intervention that the child receives, but also whether the video belongs to pre- or post-intervention groups. To do this, the videos will be sent after being encrypted and randomized for quantification. 

The study was approved by the Research Ethics Committee of the University of Murcia (1823/2018) and Comité de Ética de la Investigación de A Coruña-Ferrol (2022/099). This trial has been registered under the following code: ClinicalTrials.gov Identifier: NCT0609261.

### 2.2. Study Location

The study will be carried out in the Physical Medicine and Rehabilitation Service of the Teresa Herrera Maternal and Child Hospital of the A Coruña and Cee Health Area. Further sites may be joining the recruitment once their approvals are finalized. The study will be conducted between January 2024 and December 2024 in the Physical Medicine and Rehabilitation Service of the Teresa Herrera Maternal and Child Hospital of the A Coruña and Cee Health Area.

The data collected will pertain to subjects at risk of neurological alteration or diagnosis of cerebral palsy or neuromotor disease, who are admitted for assessment and/or physiotherapy treatment. Following the usual clinical practice protocol, the children will be assigned to one of the physiotherapists who apply Vojta therapy (patients who will be considered as the intervention group) or to a physiotherapist who applies conventional treatment (patients who will be considered as the control group).

### 2.3. Participants

Participant selection allows the randomization of the sample following the natural ordering of the rehabilitation service where the study is carried out. The assignment of children to their different therapists is performed according to the available treatment times of the latter. In other words, patients are assigned to one treatment or another following criteria of immediacy as part of the regular routine of the rehabilitation service. Therefore, the randomization will be carried out through the needs of the service, as is already the case.

#### 2.3.1. Inclusion Criteria

Male and female children between 0 and 36 months, with a risk of neurological alteration or diagnosis of cerebral palsy or neuromotor disease, who are admitted for assessment and/or treatment according to the Vojta therapy methodology or conventional treatment in the Physical and Rehabilitation Medicine Service will be included in this study. The diagnosis may be clinical and/or supported by the presence of lesions in complementary tests. Due to the fact that the diagnosis of cerebral palsy is confirmed in many cases after one year of life, the data of children in treatment for risk of neurological alteration will only be included retrospectively when said diagnosis has been confirmed. Younger children have not received a definite diagnosis of CP yet, but, based on the presence of lesions determined by magnetic resonance imaging, their risk of developing CP was estimated to be sufficiently high to justify their inclusion in the CP group. 

Parents or guardians must sign the informed consent that will allow us to use their data on therapeutic results, always in a pseudonymized (coded) way, guaranteeing confidentiality. 

#### 2.3.2. Exclusion Criteria

The data of those patients who have had to apply another therapy (pharmacological, surgical, or rehabilitative) during the care provided by the physiotherapist in their usual clinical practice will not be included. For ethical reasons, this will not be explicitly explained to the families who agree to participate in this study, so as not to influence possible therapeutic decisions in the following two months. Data of the children will be followed up and can be excluded later if there is a change in their diagnosis and cerebral palsy is not confirmed at the appropriate age. The previous therapeutic trajectory, as well as concomitant treatment (such as pharmacological treatment for spasticity, dystonia, epilepsy, etc., as well as other rehabilitation interventions) was not considered within the exclusion criteria for two reasons: their effect would already be included within the rate of motor development before VT, and also because all the children were compared only to themselves. These will be noted and considered during the calculations, but the data exclusion will only occur if there are changes in them (both adding or withdrawal) during our 2 months of intervention. 

### 2.4. Outcomes

Two of the most accepted scales in pediatric rehabilitation will be used, quantifying gross motor function from different aspects.

The first is the Gross Motor Function Measure (GMFM). Gross motor function and mobility have important roles for classification, assessment, and research involving children with neuromotor disorders [42,43,44]. The Gross Motor Function Measure [45] could be currently considered as gold standard for the quantification of gross motor function in the pediatric rehabilitation. These follow ups of thousands of children with cerebral palsy described the long-term development of their gross motor function and defined five unalterable levels of severity. By the age of six, stability over time on the prognosticated level of mobility is expected [45]. Even though the GMFM was not designed to assess outcomes of therapeutic interventions, some authors have used its centiles to compare individual effects. The use of the reference percentiles for the GMFM as outcome make it possible to estimate the change between two subsequent assessments and to directly compare gross motor improvement across ages and Gross Motor Function Classification System (GMFCS) levels. On the other hand, uncompleted items of GMFM assessments are used as short-term therapeutic goals and, therefore, individually controlled comparison of the rate of acquisition of gross motor function seems to be possible within each range. 

The Infant Motor Profile (IMP) scale [46] is another evidence-based method of assessing infant motor behavior. It not only quantifies motor milestones, but also tests movement quality by analyzing five factors: variability, adaptation, symmetry, fluency, and capacity. The advantage of this scale is that the assessment is performed through video recording, allowing us to have a dedicated clinical evaluator blinded to the type of intervention. 

### 2.5. Other Variables

Age.Diagnosis.Locomotion stage.Gross Motor Function Classification System.Previous and concomitant physiotherapy treatments.Occupational therapy.Hydrotherapy.Surgery.Botulinum toxin.Geographic background.

As we have mentioned previously, the participants in this research, within their care routine and according to the health protocols established in the Service, will be assigned to one of the following two study groups:Intervention group: patients under Vojta therapy intervention.Control group: patients under regular physiotherapy intervention.

### 2.6. Sample Size Calculation and Statistical Analysis

Sample size calculation: The prevalence of cerebral palsy is 2.08 cases/1000 births. To obtain a good sample security, 30 subjects for each of the groups to be evaluated offer a confidence interval of 95% with a security of 10.

The initial GMFM and IMP score will be divided by the child’s months of age to calculate the rate of acquisition of their items so far. This calculation represents the mean distribution of the acquisition of these motor items. The acceleration values will be used to avoid errors in the assumption of how these items were acquired, since the tendency is to be exponential at these ages. The net values obtained in the valuation will not be considered, but the rate with which they have been obtained as a function of time will be. The rate of this development without intervention will be calculated at the time of the initial assessment, relative to the age of the patient. The proportion of the score necessary to obtain in the next two months to be able to accelerate that rate, instead of following the same previous trajectory, will be an unknown concept for all therapists and examiners.

### 2.7. Intervention

Vojta therapy: The therapist applied pressure to defined zones on the body whilst positioned in prone, supine, and side-lying aspects, where the stimulus leads to automatically and involuntarily complex movement. The parents were also instructed on at least one of the exercises from the first session, after the initial assessment. The home program was progressively increased and supervised until the three therapy positions were mastered, during weekly or fortnightly follow ups. The recommended dose was four times per day at home, in session no longer than 15–20 min; however, the daily frequency of each family due to different availability was also taken into account. The frequency of the dose was divided into four groups: families who could carry on therapy (a) three times per day, (b) four times per day, (c) one or two times per day, (d) less than seven times per week or therapy at the clinic. 

Conventional physiotherapy: Conventional physiotherapy intervention included goal-directed functional training based on tasks. These motor skills will be performed in enhanced and adapted settings, but as similar as possible to the usual activities of daily living. Families and children participated in the goal setting, and the approach will focus on overcoming the limitations of the activities to reach these goals, instead of the modification of the movement patterns. This intervention is founded in motor learning and behavioral neuroscience, focusing on participation and activity acquisition. 

## 3. Methods and Timeline

Phase 1. Initial Assessment: Children who go to the rehabilitation center where they have been referred will be treated, an initial routine clinical examination will be performed, including: Personal/medical data collection.GMFM quantification.Treatment with the physiotherapist.Training of the parents in the home program.Recording for the IMP quantification.

Phase 2. Intervention period (2 months): The assignment to the therapist will be randomized according to the needs of the service, where each therapist will adapt the treatment according to the initial assessment. Both assessors and therapist cannot be blinded to the allocation or treatment, and both required live-direct performance. An external assessor (blind to both) will be able to score the IMP recordings. 

Children will receive weekly or fortnightly treatment, and their parents, family members, or carer will be trained in carrying out a home program. Each therapist will adapt the treatment according to the initial evaluation, as is usually performed in the rehabilitation service. During the treatment sessions, the physiotherapist will apply the therapy directly, either based on Vojta therapy or conventional physiotherapy. The frequency of repetition of the treatment at home and in the hospital will be recorded, depending on the availability of each family and the normal functioning of the service.

Phase 3. Second assessment at 2 months (±1 week, depending on follow up appointments).

Collection of data on the possible changes made in previous treatments and the frequency of home treatment (possible incidents due to illness or family limitations).GMFM quantification.Recording necessary for IMP quantification.

Children carried on clinically with the intervention, but this short period would allow us to attribute changes to RL, controlling better the influence of other factor (such as age maturation or other therapy inputs).

Study length. The data collection will be carried out for 4 years from the approval of the CEI. Only patients receiving healthcare with any of the therapies under investigation after authorization by the CEI will be included.

### Statistical Methods

Statistical analysis will be performed using the SPSS 27.0 statistical software for Windows (SPSS Inc., Chicago, IL, USA; version 27.0). The normality of the sample will be checked using the Shapiro–Wilk test, and if it follows a normal distribution, an analysis of variance (ANOVA) of repeated measures p will be used, with time (pre-treatment and post-treatment) as an intra-group factor, and Vojta and conventional physiotherapy groups as the inter-group factor, all with a Bonferroni correction. Statistical analysis will be performed with a confidence level of 95%, so a *p*-value of <0.05 will be considered significant.

## 4. Discussion and Ethical–Legal Aspects

Completed assessment data of four subjects has been collected to date, as two subjects withdrew prematurely or did not sign the consent form. Recruitment is currently ongoing at our first site and no results report is available due to the small sample. 

The baseline scoring of GMFM at initial assessment is related to the age of the child in order to understand the rate of items acquisition so far. The aim of this study is not to compare absolute values of GMFM between different subjects according to their age and, therefore, if the children are born preterm, the chronological age is considered. This allows us to plot changes due to spontaneous and therapeutic gross motor experience after birth. 

The rate values are calculated by dividing the total score by the time (days of age) required to achieve it, assuming that the neonatal GMFM would be zero. The acceleration factor is calculated by dividing the differences between final and initial rate (assumed to be zero at birth), by the age in days (elapsed time). The same rate and acceleration factors of acquisition are figured out between the initial assessment and after two months of the intervention. Only new points acquired during this period were considered to calculate the rate of development and acceleration values in the intervention period. Because the distribution of the item acquisition during the pre-intervention period is unknown, we performed a second rate calculation assuming that all the items were adquired during the last two months before the intervention (for children older than 2 months at initial assessment). This simulation of the maximal possible pre-intervention rate and acceleration will limit the possibility of being overcome by the post-treatment values and will avoid average values slowing down the last two preintervention months. Completing all the items in the previous two months is unlikely and would make it more difficult to obtain improvements after the intervention, especially for the older children. Even if these calculations do not benefit our results, they will ensure that the rate and acceleration post-intervention can be compared within same-lasting and following periods. We know that the GMFM item acquisition is not linear, and both calculations will help us to understand if the rate differences are due to natural acceleration or a significant therapeutic change. On the other hand, this will avoid comparing low average rate values due to longer pre-intervention periods and, therefore, random positive outcomes. Rate values within each GMFM dimension will then be considered and will help to clarify the results. 

If our hypothesis is confirmed, it would imply that gross motor function (including postural control and righting at every developmental milestone) has to be onset regardless of training and experience. Therefore, this bodily function is automatic and develops through postural ontogenesis in the first year of life. Vojta therapy would be the key to unlocking the development of gross motor function, later used in the movement of daily life activities, including other therapies, such as conventional physiotherapy, sensory stimulation, occupational therapy, etc. Data of their previous therapy inputs (physiotherapy, occupational therapy, hydrotherapy, surgery, botulin toxin) were also included, as well as Gross Motor Function Classification System (GMFCS), age, type of cerebral palsy, locomotion stadium, 21 daily doses of treatment at home, and frequency of follow ups and treatment of the therapist. The influence of these factors in our results will make us understand which interventions can have a complementary effect, as well as pointing towards the therapeutic mechanisms of VT. 

### Ethical–Legal Aspects

The development of our research will be carried out with respect to the ethical precepts established in the Declaration of Helsinki of the World Medical Association 1964 and the Convention on Human Rights and Biomedicine, made in Oviedo on 4 April 1997 and the regulations in force in terms of health, research, and protection of personal data both European, and state and regional, as applicable to our study.

The researchers participating in this study undertake that all data collected from the study subjects be separated from personal identification data, guaranteeing the confidentiality of the research participants, and respecting the European Data Protection Regulation and Spanish regulations (both state and regional) in terms of data protection, health, and research, in force and applicable to the specific case. 

Patient data will be collected in the study-specific data collection notebook (CRD), which will be duly coded (pseudonymized) by the clinical investigators belonging to the health institution. The principal investigator outside the health institution will only receive pseudonymized data and will not have access to the identification data of the participants.

Only the research team and the health authorities who have a duty to maintain confidentiality will have access to all the data collected for the study.

Only information that cannot be identified may be transmitted to third parties (the code file will not be sent, only non-identifiable data).

Once the study is finished, the data will be eliminated or kept pseudonymized for use in future research similar to the one proposed here, according to what the participant indicates in the informed consent that must be signed.

## Data Availability

Data are contained within the article.

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
