# Peer review of "Effects of Vojta Therapy on the Motor Function of Children with Neuromotor Disorders: Study Protocol for a Randomized Controlled Trial"

_jcm, 2023, doi:10.3390/jcm12237373_

Round 1

Reviewer 1 Report

Comments and Suggestions for Authors

This is proposal for future study [Effects of Vojta Therapy on the motor function of children with 2 neuromotor disorders: study protocol for a randomised con-3 trolled trial]

It is interesting topic ,objectives and methods as well as the outcome are clear .But can you add which type of cerebral palsy that will be included in the study , what about the level of cognitive /intelligent Q that will be included  . Also it is not clear if those infants will be under medication for treatment of spasticity  or etc....

Author Response

This is proposal for future study [Effects of Vojta Therapy on the motor function of children with 2 neuromotor disorders: study protocol for a randomised con-3 trolled trial]

It is interesting topic ,objectives and methods as well as the outcome are clear .But can you add which type of cerebral palsy that will be included in the study , what about the level of cognitive /intelligent Q that will be included  . Also it is not clear if those infants will be under medication for treatment of spasticity  or etc....

Dear Editor and reviewers of the manuscript entitled “Effects of Vojta Therapy on the motor function of children with  neuromotor disorders: study protocol for a randomised con-3 trolled trial”. First of all, we would like to thank you for your comments and for allowing us to address the issues you raised to improve the manuscript’s quality. We appreciate your observations and the time devoted to the constructive criticism and feedback of our manuscript. Please find the answer to your comments below and the recommended changes have been highlighted in blue in the manuscript.

We intentionally included younger children with cerebral palsy in our study in order to reduce the interaction with other treatments. This is the case for spasticity and other pharmacological treatment, but also other interventions like occupational therapy, conductive education, movement groups, etc. In any of those cases, all the interventions that each individual have at initial assessment or at the final evaluation will be noted. If there are other interventions added during our treatment, the data from this subject will not be added, as the results cannot be attributed to Vojta Therapy. If the treatment has not changed, this will not be a problem for the calculations, as the children are only compared with themselves pre-post. Ultimately, this other treatment will be considered in the calculations to understand if, for instance, children who were in treatment with VT+ spasticity medication did better than those only with VT. In both cases and due to ethical reasons, differences between children will be noted but not modified.

This has now been clarified in the text with the following paragraph at the end of the exclusion criteria section: “The previous therapeutic trajectory, as well as concomitant treatment (such as pharmacological treatment for spasticity, dystonia, epilepsy, etc, as well as other rehabilitation interventions) was not considered within the exclusion criteria for two reasons: their effect would be already included within the rate of motor development before VT, and also because all the children were compared only to themselves. These will be noted and considered during the calculations, but the date exclusion will only occur if there are changes on them (both adding or withdraw) during our 2 months of intervention.”

Regarding the types of cerebral palsy, this will also be noted. Nevertheless, because of the young age of these children, a clear diagnosis may not be possible yet. Once they finish their intervention period, they will be followed up to an age where this diagnosis could be confirmed (about 2 years old), and also the type of cerebral palsy will be clearer, and then register. Therefore, any type of cerebral palsy will be collected, initially proven for a brain injury found in imaging testing.  Similarly, to medication and other treatments, this will be considered in the calculations to understand which types of cerebral palsy responded better to the intervention.

Similarities with the cognitive level will be applied. Cognitive assessment in younger ages is based on motor development, as well as other areas such as social and language performance. Because the children will be compared only with themselves, and only after 2 months of intervention, no dramatic changes are expected in other areas other than gross motor function. 

Reviewer 2 Report

Comments and Suggestions for Authors

Only comment:

It seems really important to me that the authors quickly explain the effect of cerebral palsy on the motor behavior of children.

I am thinking here in particular of the work of Wallard et al.

- Wallard, L., Dietrich, G., Kerlirzin, Y., & Bredin, J. (2014). Balance control in cheerful children with cerebral palsy. Gait & Posture, 40(1), 43-47.

or the work of Pierret et al.
- Pierret, J., Beyaert, C., Vasa, R., Rumilly, E., Paysant, J., & Caudron, S. (2023). Rehabilitation of Postural Control and Gait in Children with Cerebral Palsy: the Beneficial Effects of Trunk-Focused Postural Activities. Developmental Neurorehabilitation, 26(3), 180-192.

All of these work have highlighted the control of balance during children's locomotion.

Author Response

Only comment:

It seems really important to me that the authors quickly explain the effect of cerebral palsy on the motor behavior of children.

I am thinking here in particular of the work of Wallard et al.

- Wallard, L., Dietrich, G., Kerlirzin, Y., & Bredin, J. (2014). Balance control in cheerful children with cerebral palsy. Gait & Posture, 40(1), 43-47.

or the work of Pierret et al.

- Pierret, J., Beyaert, C., Vasa, R., Rumilly, E., Paysant, J., & Caudron, S. (2023). Rehabilitation of Postural Control and Gait in Children with Cerebral Palsy: the Beneficial Effects of Trunk-Focused Postural Activities. Developmental Neurorehabilitation, 26(3), 180-192.

All of these work have highlighted the control of balance during children's locomotion.

Dear Editor and reviewers of the manuscript entitled “Effects of Vojta Therapy on the motor function of children with neuromotor disorders: study protocol for a randomised con-3 trolled trial”.

First of all, we would like to thank you for your comments and for allowing us to address the issues you raised to improve the manuscript’s quality. We appreciate your observations and the time devoted to the constructive criticism and feedback of our manuscript. Please find the answer to your comments below and the recommended changes have been highlighted in blue in the manuscript.

We followed your advice and included in the text the effect of cerebral palsy in the motor function, according to the articles given. The following text has been added in the introduction section:

“Children with CP have specific deficits of postural control in standing or walking, including reduced stability of the axial segments. This results in a set of persistent atypical movements and posture, which prevent balance, as well as fluid and selective movements. Compensatory postural strategies described as "on bloc movements” are often observed, for instance, non-selective swinging between head and trunk as a single segment to achieve forward propulsion without losing balance [17]. These compensatory strategies must be identified and considered in rehabilitation programs for CP children. These aspects may be improved with rehabilitative interventions focusing on the postural control and trunk activity [18,19]. “

Reviewer 3 Report

Comments and Suggestions for Authors

I read this manuscript and found it of certain interest. However, it needs revision:

1.     Abstract: It needs more information

2.     Introduction: It would be immensely beneficial for clinicians if you could dive deeper into the topic of "Vojta Therapy" and provide some historical context or previous studies involving its application in pediatric cases.

3.     Methods: I must commend you on the study design; it's quite solid and seems entirely acceptable. However, I do have reservations about the relatively diminutive sample size of 30 patients. The patient groups appear somewhat meager in numbers, and I'm left wondering about the robustness of the statistical power calculations that led to this choice. The evaluations conducted on these small groups could potentially benefit from further clarification and consistency.

4.     It would be greatly appreciated if you could elucidate the criteria for selecting therapy for patients, whether it be Vojta Therapy or the conventional approach. Clarity in this aspect would go a long way in enhancing the manuscript.

Comments on the Quality of English Language

The English of the manuscript needs revision. There are some grammatical and syntax errors in the manuscript

Author Response

Dear Editor and reviewers of the manuscript entitled “Effects of Vojta Therapy on the motor function of children with  neuromotor disorders: study protocol for a randomised con-3 trolled trial”.

First of all, we would like to thank you for your comments and for allowing us to address the issues you raised to improve the manuscript’s quality. We appreciate your observations and the time devoted to the constructive criticism and feedback of our manuscript. Please find the answer to your comments below and the recommended changes have been highlighted in blue in the manuscript.

  1. Abstract: It needs more information

The abstract information has now modified as requested.

  1. Introduction: It would be immensely beneficial for clinicians if you could dive deeper into the topic of "Vojta Therapy" and provide some historical context or previous studies involving its application in pediatric cases.

We have now added a wider research background, although this is limited because the transmission of classical therapies has been traditionally done as apprentice training. This is the main reason backing up our research, offering the most accurate possible design to understand if Vojta Therapy has any or none effects in children with cerebral palsy.

“VT was developed by the Czech professor, pediatric and adult neurologist, Vaclav Vojta in the 60´s. Most of his rehabilitation work and the development of the Vojta Principle was developed in Germany where he emigrated due to the political situation in Europe after the Second World War. The physiotherapy developed by Professor Vojta using “reflex locomotion” is based on his findings on the development of posture and movement in children. He discovered the basis of this therapeutic intervention while searching for treatment possibilities for young people with CP and spastic paralysis [7]. His approach was founded in years of systematic neurokinesiological observation and examination of children with typical and atypical development. He observed that sustained stimulation of peripheral pressure elicited a stereotyped full-body motor response, manifested as a pattern of tonic and phasic muscle contractions on both sides of the neck, trunk, and limbs resulting from spatial and temporal summation that leads to improved postural control [6].”

“Studies using EEG show changes in motor areas, indicating the activation of innate locomotor circuits [12].”

“Numerous studies have reported improvements in the acquisition of sitting in infantile cerebral palsy [18], gait acquisition [19,20], improving gait kinetics and kinematics [21–23], as well as stability, trunk balance and spinal asymmetry [24,25]. Furthermore, Vojta Therapy in infantile cerebral palsy is considered one of the most recommended techniques for the treatment of respiratory disorders in infantile cerebral palsy [26] as it has proven to be the technique of choice in respiratory physiotherapy as it has been shown to improve compliance, SO2, SpO2 by reducing the respiratory frequency without increasing PCO2 [27–29]”

  1. Methods: I must commend you on the study design; it's quite solid and seems entirely acceptable. However, I do have reservations about the relatively diminutive sample size of 30 patients. The patient groups appear somewhat meager in numbers, and I'm left wondering about the robustness of the statistical power calculations that led to this choice. The evaluations conducted on these small groups could potentially benefit from further clarification and consistency.

Thank you for your support with our design. We are aware of the limitations of this type of work, and we tried to solve it with the most solid and realist design possible.  We agree that 30 subjects are a limiting factor, however they seemed to be a realistic target for each interventional site (we currently have only one). We are currently recruiting more sites to increase the sample size, but a precise protocol will be needed to guarantee the reliability among them. Peer reviewing is part of this process, and we will update this number once new sites have been confirmed.

  1. It would be greatly appreciated if you could elucidate the criteria for selecting therapy for patients, whether it be Vojta Therapy or the conventional approach. Clarity in this aspect would go a long way in enhancing the manuscript.

We have now better described the allocation of the children to the different interventional groups in the sentence below and integrated within the method section in “blue”. Pure randomized allocation will not be an ethical design, so we used the pseudo-randomization occurring regularly due to the service needs and consultant criteria. The children will be allocated to different physiotherapists depending on the availability, immediacy, and rehabilitation consultant criteria for treatment prescription. This is the normal functioning of the rehabilitation service, so this design does not require an unethical modification, but provides a service evaluation.

Please see in the text:

“The ethical limitations of randomized untreated control groups or placebo interventions limit the quality of the intervention in this field.”

“A pseudo-randomized clinical trial has been designed that will assess gross motor function before the physiotherapy intervention (with Vojta therapy or conventional physiotherapy intervention as environmental enrichment and task specific training”

Patients are assigned to one treatment or another following criteria of immediacy as part of the regular routine of the rehabilitation service. Therefore, the randomization of the sample is possible following the natural ordering of the rehabilitation service where the study is carried out. The participants in this research, within their care routine and ac-cording to the health protocols established in the service, will be assigned to one of the two study groups:

  • Intervention group: patients under Vojta Therapy intervention (17-29)
  • Control group: patients under regular physiotherapy intervention as environmental enrichment and task specific training (26)

This methodology will be useful to provide a service evaluation on the operation of the rehabilitation service itself.”

We have also described the type of “conventional physiotherapy” provided from the non-VT therapist:

“Conventional physiotherapy: Conventional physiotherapy intervention included goal-directed functional training based on tasks. These motor skills will be performed in enhanced and adapted settings, but as similar as possible to the usual activities of daily living. Family and children participated in the goal setting, and the approach will focus on overcoming the limitations of the activities to reach these, instead of the modification of the movement patterns. This intervention is founded in motor learning and behavioral neuroscience, focusing on participation and activity acquisition.”